# ComSum: Commit Messages Summarization and Meaning Preservation

**Leshem Choshen**[*]
Department of Computer Science
Hebrew University of Jerusalem
leshem.choshen@mail.huji.ac.il

**Idan Amit**[*]
Department of Computer Science
Hebrew University of Jerusalem and Acumen Labs
idan.amit@mail.huji.ac.il

## Abstract

We present ComSum, a data set of 7 million commit messages for text summarization. When documenting commits, software code changes, both a message and its summary are posted. We gather and filter those to curate developers' work summarization data set. Along with its growing size, practicality and challenging language domain, the data set benefits from the living field of empirical software engineering. As commits follow a typology, we propose to not only evaluate outputs by Rouge, but by their meaning preservation.

## 1 Introduction

There is an ever-growing amount of code written in the world. When code is created by large groups of developers, documentation becomes essential. As a part of it, developers' proper documentation is also related to code quality [Santos and Hindle, 2016]. The need to communicate is especially important in distributed development, where shouting over the hallway cannot compensate for improper documentation.

Code development nowadays is usually supported by version control systems that track the source code modification. The most common such version control system is Git. In Git, each modification is called a *commit*. A commit lists the changed lines in the source code and a description by the developer. The description contains a one-line subject and a longer message describing the commit. Git and GitHub treat the subject line as a summary[2], further incentivizing developers to do the same. Hence, in order to build a text summarization data set, we use the subject as the summary and the rest of the commit message as the source.

In Section §3, we describe the process of querying and filtering commits to curate ComSum, a commit summarization data set. The data set is described in Section §4. We consider several baseline results on the data set (see Section §5). The baselines include both neural summarization baselines and baselines based on relations between the subject and the message. Those shed light on the state of the art on the data set and the data set characteristics.

Since commits are used to describe code changes, the taxonomy of changes is an important part of their meaning. That enables us to evaluate a summary model by how well it preserves the meaning rather than by word overlap with a reference. We explain and provide initial analysis in Section §6.

Submitted to the 35th Conference on Neural Information Processing Systems (NeurIPS 2021) Track on Datasets and Benchmarks. Do not distribute.

| Dataset | # Docs. | Coverage | Density | Comp. Ratio |
|---|---|---|---|---|
| Arxiv/PubMed [Cohan et al., 2018] | 346,187 | 0.87 | 3.94 | 31.17 |
| BigPatent [Sharma et al., 2019] | 1,341,306 | 0.86 | 2.38 | 36.84 |
| CNN/DM [Nallapati et al., 2016] | 311,971 | 0.85 | 3.47 | 14.89 |
| Newsroom [Grusky et al., 2018] | 1,212,739 | 0.83 | 9.51 | 43.64 |
| XSum [Narayan et al., 2018] | 226,677 | 0.66 | 1.09 | 19.25 |
| BOOKSUM Paragraph [Kryściński et al., 2021] | 142,753 | 0.5 | 0.92 | 6.47 |
| ComSum | 7,540,026 | 0.27 | 0.89 | 8.1 |

Table 1: ComSum is more abstractive, as seen in low coverage and density.

## 2 Related Work

Textual Summarization has become a central task in Natural Language Processing, attracting pre-trained models [Zhang et al., 2020, Qi et al., 2020] and specialized models [Dou et al., 2021]. With this came also a need for larger, more diverse and more challenging data sets.

News is the most common domain for summarization data sets [Over et al., 2007, Nallapati et al., 2016]. Narayan et al. [2018] proposed a data set under the motivation of creating a large news data set which does not favor copying the source and extractive summarization. In parallel, Grusky et al. [2018] proposed an even larger data set for news extractive summarization. While news is advocated for its general domain, we find the vocabulary which should demonstrate it is rather low in comparison to our domain. The vocabulary of the commits is over 2M in the validation set alone (to be fair in terms of size) and 19M overall (top reported is 1.4M NYTimes dataset[Narayan et al., 2018, Sandhaus, 2008]). Similarly, the vocabulary of the summaries is 0.5M and 3.9M (NYTimes 0.3M).

Kryściński et al. [2021] called for more challenging and diverse abstractive summarization data sets, releasing a long narrative summarization data set providing several versions with 143K examples at the largest one. We compare (Table 1) the abstractness as measured by low density and coverage [Grusky et al., 2018].

Others offered large datasets of different domains 4M crawled TL;DR from reddit [Völske et al., 2017] and 1.3M patents [Sharma et al., 2019].

Our work follows all those desirable traits. It is more abstractive, it introduces a new natural summarization domain, it is even larger than current data sets and it is expected to keep growing in size substantially.

Several data sets and tasks share similarities with summarization. Those include simplification [Alva-Manchego et al., 2020], sentence compression [Filippova and Altun, 2013], web-page snippet generation by query [Chen et al., 2020] and single sentence summarization [Rush et al., 2015].

Evaluation of summarization mainly focuses on general quality of summarizations [Bhandari et al., 2020, Zhang et al., 2019], with some exceptions [Wilber et al., 2021, Xu and Durrett, 2021]. Some work showed hallucinations are a problem [Kryscinski et al., 2020] and focused on evaluation of factual consistency [Gabriel et al., 2020, Honovich et al., 2021, Pagnoni et al., 2021]. Other fields of text generation provide additional ways to extract informative measures [Ribeiro et al., 2020]. Measures that tell about certain characteristics of the output, rather than bottom-line scores. Such methods include evaluation on minimal changes to the input [Warstadt et al., 2020], challenge sets [Macketanz et al., 2018, Choshen and Abend, 2019], metrics dedicated to specific characteristics such as grammaticality [Vadlapudi and Katragadda, 2010] or meaning preservation [Choshen and Abend, 2018c], manual evaluation [Graham et al., 2015], evaluation dependent on the domain data [Choshen and Abend, 2018a], understanding the inner workings of networks [Tenney et al., 2019, Slobodkin et al., 2021, Voita et al., 2020] and more. In addition to the data set, we propose (see Section §6) an evaluation procedure specific to the domain at hand that emphasizes the overall meaning of the summary rather than its similarity to a reference.

---

*First two authors contributed equally.

[2]See for example here: `https://github.com/tensorflow/tensorflow/commits/master`

Apart from the summarization aspects, this work also contributes to the growing field of aiding programming through NLP, including a dedicated workshop [3]. This field includes tasks such as code generation [Hayati et al., 2018], automatic documentation [Miceli Barone and Sennrich, 2017], code search [Gu et al., 2018] and updating documentation by code changes [Panthaplackel et al., 2020].

# 3 Data Set Creation

In this section, we describe the creation of the proposed data set, ComSum. Specifically, we describe how we acquire and filter projects and commits to extract reliable summarizations of messages from the subjects.

## 3.1 Data acquisition

Open source code is shared in large amounts on different platforms. GitHub, owned by Microsoft, is a large hosting service for projects using the Git version control system. In 2018 GitHub published that they hosted 100 million projects[4].

We base our data set on the BigQuery GitHub schema[5]. The schema allows querying commits pushed to GitHub by various metadata attributes. The BigQuery GitHub schema contains about 3.4 million *public* projects prior to 2021, but the vast majority are not appropriate for studies of software engineering, being small, non-recent, or not even code.

## 3.2 Projects selection

Code is written in the context of repositories or projects, sharing a general goal and mostly developed by the same people or groups. While some projects consist of a few files shared, others are constantly updating. We focus on the latter, aiming to filter the first to avoid commits where clear communication is neither crucial nor enforced.

Therefore, the main effort in the data set construction is to identify projects of interest and filter irrelevant ones. We base our filtering on the methodology developed by Amit and Feitelson [2020] to study software code quality. First, we choose only large enough and up to date projects by requiring at least 50 commits during 2020[6]. Note that this is less than one commit per week, a rather low bound filtering tiny projects. However, this step alone was enough to reduce the number of relevant projects to 32,562, 0.96% of the prior step.

The next step is the removal of redundant projects. Github enables 'forking': copying a project, sometimes for investigation and sometimes for modifying without altering the main project. We identified forks using the GitHub API and removed them from our data set. We also removed projects that shared too many commits with a more popular project, having more stars, in order to use 'Spark' of the 'Apache Software Foundation' and not a hobbyist project built upon it. This step ended with 25,535 projects, 78% of the prior step.

We filtered projects not likely to be software projects, identified by the lack of bugs. A project without commits fixing bugs was identified by a negative Corrective Commit Probability (CCP) [Amit and Feitelson, 2020]. Projects with negative CCP are less likely to be software projects, externally validated by programming languages identification and imply no use of the pull request development mechanism. After filtering projects with invalid CCP we were left with 22,820 projects, 89% of the previous step. Out of these, 19,720 projects had commits fitting the conditions described next, 86% of the previous step and 0.57% of overall projects.

---

[3]https://nlp4prog.github.io/2021/cfp/

[4]https://github.blog/2018-11-08-100m-repos/

[5]`https://console.cloud.google.com/marketplace/product/github/github-repos`

[6]Future projects requiring more data or updated data should consider keeping the requirement of 50 commits per year, but setting a different set of years instead of the singleton (2020). Also, consider changing other filters, specifically the 100 characters difference.

### 3.3 Commits Selection

When constructing a data set, the intended use is of high importance. In our case, many future uses are possible. One might want to remove single-person projects since they do not represent communication. Others might be interested only in single person projects since they represent documentation for a future self. We choose not to add needless filtering here in order to allow future flexibility.

We used only two constraints. We require that the message will be at least 100 characters longer than the subject. While the value of 100 is arbitrary, a significant gap in the length is required in order to have a meaningful summarization.

We considered only commits earlier than 2021 to help future reproducibility. Note that this condition is not enough to guarantee the stability of the data set since existing projects might be deleted and their content will be removed from the index. In order to cope with that, we release a static version of the data set together with the code to extract it.

Overall, the data set has 7,540,026 commits. A commit might appear in a few repositories, having the same subject, message and content. Since we are interested in text summarization, those are considered duplicates and we remove those 16% of the data that was repetitive. This step is especially important to prevent training data leaking to the test set.

A message or a subject might appear more than once. For example, the most common subject is "Updating submodules", appearing 6612 times. However, 96% of the subjects are unique. We left the multiple appearing message since this represents software development. Researchers looking for uniqueness can remove multiple appearing texts. We provide appearance distribution and common messages.

In 0.8% of the cases the subject appeared in the rest of the message. We extract the common ones and they seem to be due to a generic subject. The leading ones are 'WebCore:' (1554 times), 'Updated Spanish translation.' (809 times), 'Updated Norwegian bokmål translation.' (347 times). Again, in order to enable maximal future flexibility we left them. An interested researcher can filter these few commits out.

Another question of interest is to what extent, the subject represent a summary of the message. Our labeling protocol is described in the supplementary materials. In essence, we require a proper summary to add no new information and to contain the gist. We also labeled whether the summary was generic or specific to the commit.

We manually labeled 100 samples. In 80% the subject was a summary of the message. Out of the rest, 20%: 35% had administrative message (e.g., just the reviewer detail). 20% had a subject which indicates a merge (e.g., merge branch 3.4) and the message as the content. In 15% the subject was by reference (e.g., fixed #123) In 5%, there was a generic subject. The rest of the 25% are diverse and harder to identify and handle.

We provide a list of 429K merge commits (identified by having more then one commit parent, might have informative subject) to enable to remove them.

We also provide a *heuristic* for administrative messages. We identify them by the distance of administrative terms (e.g., 'Signed-off-by:', 'Change-Id:') from the beginning of the message. Manual labeling (See App. §B) shows those to have 98.9% precision and 75% recall. We didn't filter these commits, allowing researchers to change the distance threshold and trade off recall and precision.

Using both filtering leads to about 90% of the subjects serving as summaries on our labeled sample.

## 4 Data Set Description

Following all the above procedures we create ComSum, a data set of commit messages and their summarization. ComSum contains 7,540,026 commits from 19,720 projects, written by 317,423 authors.

In addition to the summarization objective, we publish metadata on each commit. Each record also has the commit hash, a unique identifier that can be used for future work. Last, we provide the name of the repository from which the commit was taken. We note that a commit might be included in several repositories (projects), but we choose only one (see Section §3).

Our guideline in the data set construction was to ease consistent use by researchers. Hence, each record contains the subject, the message without the subject and the entire commit message combining the two. While it almost doubles the storage, it prevents future noise due to different implementations.

The average subject has 53 characters, the average message has 494 characters. The average compression rate, the commit message length divided by its subject length is 11.08 .

We separate the data set into train, validation and test in order to compare future results in a consistent way. The separation is based on the commit hash so it is both pseudo-random and reproducible. The test set and the validation set have about 472K examples each. Hence, a project may appear in both the train and the test, and so does the same summary. However, a message and its summary never repeats.

Overall, 418,994 (89%) subject lines from the test set lines never appear in the training set. Other lines are common and appear repeatedly in the train set (e.g., Merge branch 2.7 into 2.8). However, manual inspection suggests their corresponding commit messages share little resemblance. As the model should learn when to recall more generic summaries as well as to generate specific ones well, we leave those in the test set and do not separate train and test by subjects. We also create subsets of meaning-preserving commits, explained in Section §6.

## 5 Baselines

We inspect the behavior of neural models and baselines. Those provide insight on the characteristics of the data set, set a baseline for future work and allow us to consider unique evaluation motivated by domain knowledge (see Section §6). In all experiments, we compute the Rouge1, Rouge2 and RougeL scores [Lin, 2004].

For a neural model we used BART [Lewis et al., 2020]. We consider two variations of BART, one untrained for zero-shot performance and another fine-tuned on the train data set. We used max target length of 128 and source length of 512, learning rate of $1e^{-4}$ and 256 batch size. The rest of the parameters are the defaults by the HuggingFace library. The model was trained for a week on 4 Nvidia M60 GPUs.

BART is originally trained in the domains of Wikipedia and Books and it was not exposed to the non-formal and technical language found in commit messages. On the other hand, BART, pre-trained that way, showed impressive results even on very far domains such as malware detection based on dynamic analysis [Oak et al., 2019]. Anyway, BART results are high (Table 2) and it surpasses Zero-shot results by a large margin, suggesting the data set is large enough to overcome the domain shift at least partially. Lewis et al. [2020] reported that BART achieved RougeL of 44.2 on the CNN and Daily Mail data sets [Hermann et al., 2015] and 37.6 on the XSum data set [Narayan et al., 2018], better results than on ComSum.

Following previous work [Kryściński et al., 2021], we provide heuristic summarization techniques for analysis. Results were computed on 10k samples and presented in Table 2. These heuristic summarizing techniques do not learn and therefore the train and test splits are irrelevant to them.

As a first, 'Subject and Message' do not summarize at all and include both the subject and the message, acquiring a 29.5 RougeL. This demonstrates the need for compression. Note that this method cannot be used for prediction since it uses the summary.

We perform another such test using the *Message without Subject* as the summarization. This method reached a RougeL score of 12.3 which is better than other baselines but worse than zero-shot performance of BART. This implies that while there is information in the commit message, repeating it is far from enough to achieve a good summarization score in ComSum.

Similarly, we define a *Random Message Sentence* baseline. We split the message into sentences and randomly pick a single non-empty sentence as the summarization, which achieves a RougeL of 12.4. This comes to see how well a more reasonably sized extraction of the input would do (on average, messages are 11.08 times longer than subject). As expected it is worse than the whole message and shows sentence extraction is unlikely to be considered a good commit summarization.

| Model | Data set | RougeL | Rouge1 | Rouge2 |
|---|---|---|---|---|
| Bart | Train | 36.6 | 38.5 | 22.1 |
| Bart | Test | 36.3 | 38.2 | 21.8 |
| Zero-Shot Bart | Train | 17.8 | 20 | 8.2 |
| Zero-Shot Bart | Test | 17.9 | 20 | 8.3 |
| Subject and Message | All | 36.7 | 36.7 | 34.5 |
| Message without Subject | All | 15.2 | 18.0 | 8.3 |
| Related Fix | All | 14.9 | 17.4 | 8.6 |
| Random Message Sentence | Train | 12.4 | 13.9 | 5.8 |
| Random Message Sentence | Test | 12.4 | 13.8 | 6.0 |
| Related Commit | All | 7.6 | 8.0 | 3.4 |

Table 2: Baselines results on different data sets. Training on the data set provides a significant boost. Repeating the commit message or a related subject is not enough for a high score.

Another test case is *Related Commit*. We generate pairs of commits by the same author, in the same project, within a week's range of each other. We consider the subject of one commit as the summary of its paired message, mimicking a situation in which the summarizing is at the level of the same person speaking on the same topic, regardless of the specific message. We expect high scores from such a measure if subjects are general and quite similar or even repetitive upon related commits. The related commit subject benchmark reached the score of 14.4 , suggesting this is not the case. Where we require both commits to be a bug fix, a setting we term *Related Fix* the results are higher. Results are also somewhat higher than those achieved by extracting a 'Random Message Sentence' from the commit. This shows that the topic and style conserve some of the meaning needed for the summary, but they are far from satisfactory surrogates of a summary. Please note that in the text summarizing we treat each commit on its own and compare the commit message and subject. We use more than one commit here only as a predictor for benchmark.

Memorization is both a sign of over-fitting and a known model behavior in some cases [Feldman, 2020]. A way to evaluate its influence is to compare the performance on the train and test sets Arpit et al. [2017]. It appears that memorization is not a strong problem as both BART and Zero-Shot Bart results on the train and test are quite similar (and the sets do not contain duplicates).

Surprisingly, BART achieves similar results to that of the message and subject, which is a summarization that includes all needed summary (the actual reference) and a lot of unneeded text.

Manually inspecting the outputs of BART shows mixed results. On the one hand, a reasonable percentage of sentences resemble the reference and in general convey the right action done in the commit. On the other hand, many errors are found. Even well-structured sentences fail in terms of factual consistency. The network hallucinates terms, names and numbers. For example, "Merge pull request **#14**" instead of #1110, "Bump flake8-isort from 2.9.0 to **2.8.1**" instead of to 2.9.1 and other more complex cases. The high Rouge score co-existing with common mistakes suggest that other evaluation procedures should be suggested to differentiate allowed sentence variants from outputs of wrong meaning.

# 6 Meaning Preserving Summarization

Meaning preserving is part of the definition of the text summarization problem Gupta and Lehal [2010], Chopra et al. [2016]. Gupta and Lehal [2010] suggested an elegant mathematical definition $model\ summary = \text{argmax}_x\ p(message|x)$.

While attractive, this definition suffers from several drawbacks. The choice of the probability model is subjective. The computation of the most likely summary might not be feasible. Last, it is not clear if the writer intends and is capable of summarising the message that way, meaning that the samples that we use do not fit the concept aimed to learn.

Testing summarization quality by word overlap alone might be unreliable [Choshen and Abend, 2018b] and human annotation is costly. Fine-grained factual consistency is less specific to this domain and is an active field of study [Gabriel et al., 2020, Honovich et al., 2021]. We hence, provide specialized test approaches, asserting that the output summaries preserve the meaning.

There are many aspects of meaning that one might choose to preserve. For example, the sentiment of the author, the register of the language, etc. A good aspect to preserve should have a few properties.

First, there should be an agreement that this meaning should be preserved. Given a model that does not preserve sentiment, one might claim that this is desirable, leading to a more concise summary removing irrelevant information.

The second property should be that the aspect can be estimated using a computable function, a requirement for automation on a large scale. The aspect should be as objective as possible, (e.g., as measured by agreement between human annotators), in order to avoid a model that has a different subjective "point of view".

Our data set enables the use of the commit type, having these properties. We use the classical commit taxonomy of Swanson, suggested in 1976, classifying commits as: corrective (aka bug fix), adaptive (adding new features) and perfective (refactoring and documentation improvements) [Swanson, 1976]. This taxonomy is very common among developers and software engineering researchers [Mockus and Votta, 2000]. Therefore we used a model for corrective, adaptive and refactor, a subset of perfective. We chose to focus on the latter as refactor changes are related to the source code and are therefore more important to communicate. The classification captures the essence of the work done in the commit, hence, its meaning should be preserved.

A commit might be tangled and serve several goals, for example, both fix a bug and refactor the fixed code [Herzig and Zeller, 2013, Herbold et al., 2020]. Other than being less common, being a bug does not influence being a refactor and both meanings should be preserved.

Human annotators reach an agreement of 95% on the classification of a commit as a bug [Amit and Feitelson, 2020]. We use the classifiers of [Amit and Feitelson, 2019, 2020], reaching accuracy of 93% for corrective and refactoring , very close to the human level, and the adaptive classifier of accuracy 65%. Hence we are capable of estimating the classification at scale accurately.

One could use the classification on random commits as the meaning to preserve. However, a naive model identifying a list of core terms, whose appearance is indicative of the concept, like 'bug', 'bug-fix', 'error', 'fail', and 'fix' reaches an accuracy of 88% classifying the corrective concept. Since these words are common in commit messages, a model ignorant of the meaning might still use them as a summary. Therefore, we suggest a more challenging cases for meaning preservations analysis.

We compute for all the discussed concepts the precision-like meaning-preservation metric, $P(concept(model(message)))|P(concept(message)$. BART's highest precision for any of the concepts we tested on was 75%. This emphasizes how common and severe non-preserving summaries are and it calls for further investigation. However, an alternative claim is that omitting the concept from the summarization is fine since it is not important. In order to cope with this claim, we construct cases in which the classification as the concept is added and not omitted.

A naive model will fail on sentences like "Added *error* handling", "Used *fixed* point arithmetic", "This is not a *bug fix* but a new requirement", etc. In order to build a suitable data set, we selected messages that contain a core term yet classified as negative by the concept's classifier. I.e., they contain a core term that usually suggests they belong to one concept, but they do not.

Hence, in order to preserve the meaning, the summary should match the message in concept. In that case, the system output should be of the negative class too and preferably contain the core term.

Before evaluating meaning preservation, we present the Rouge score on the meaning preserving data sets. Comparing the results in Table 2 and Table 3, shows similar performance trends.

However, the meaning-preserving property allows us to extend our analysis beyond this bottom line. Table 4 presents the classification of the summaries of the meaning-preserving messages that have a core term of a concept yet are not part of the concept's class. Such a message might be "Added *error* handling" that is not classified as a bug fix despite the appearance of the core term "error". When a message contains a core term but is still classified as having a concept, it indicates that the concept is indeed not the core's one, as the prior on matching concept and core term is very high. We build such data sets for corrective, refactor and adaptive concepts in order to demonstrate it is not a property of a specific concept.

Next, we observe the summaries generated by Bart. When the summaries include a core term, yet are not classified as discussing a concept, the meaning is correct, matching the message. This is the

| Model | Data set | RougeL | Rouge1 | Rouge2 |
|---|---|---|---|---|
| Bart | Adaptive | 36.3 | 38.4 | 21.5 |
| Bart | Refactor | 36.1 | 38.0 | 22.2 |
| Bart | Corrective | 36.8 | 38.6 | 22.2 |
| Zero-Shot Bart | Adaptive | 18.6 | 20.5 | 8.7 |
| Zero-Shot Bart | Refactor | 18.5 | 20.5 | 8.5 |
| Zero-Shot Bart | Corrective | 18.8 | 20.8 | 9.4 |
| Random Message Sentence | Adaptive | 12.4 | 13.9 | 5.5 |
| Random Message Sentence | Refactor | 11.4 | 12.8 | 4.8 |
| Random Message Sentence | Corrective | 12.3 | 13.9 | 6.2 |

Table 3: Rouge scores on typed test sets. Trends are similar to those on the general test set.

| Model | Data set | Not Preserved | Core and Concept | Not Core and Concept | Core and Not Concept | Not Core and Not Concept |
|---|---|---|---|---|---|---|
| Bart | Corrective | **0.21** | 0.16 | 0.04 | 0.28 | 0.52 |
| Bart | Refactor | **0.11** | 0.07 | 0.04 | 0.11 | 0.78 |
| Bart | Adaptive | **0.39** | 0.27 | 0.12 | 0.19 | 0.42 |

Table 4: Meaning Preserving on summaries containing a distractor core term (e.g., "bug") not fitting their concept type (e.g., corrective). Models are more likely to preserve the core term than the meaning. Furthermore. those cases are confusing for the model.

best case where the summary matches both the concept and the core term. Optimally, all summaries would fall under this case.

When there is no core term and the summary is not classified as a (wrong) concept, it might be a good summary in terms of meaning, not stating the nonexistent concept. On the other hand, these cases might be a result of hallucinations, as they do not mention the core term.

However, when there is a core term and the summary is classified as the concept, then the meaning was changed. Last, when there is no core term and the message is classified as discussing the concept, not only the meaning is changed, the core term disappears and the summary might be a result of hallucinations too. "Not Preserved" in the table represents the cases where the meaning was changed. It is the sum of "Core and Concept" and "Not Core and Concept". We find that 11-39% of sentences checked change their meaning. These meaning-preserving probabilities serve as quality metrics for the summary. Future work may integrate them into the loss function, forcing the model to both produce the right words and keep the meaning.

It is worth reiterating that the commit classifiers are not perfect. While they were trained on thousands of messages, the meaning preserving messages are different in nature and the distracting core term makes them harder to classify. We manually labeled 20 messages for each concept, in order to estimate how many of the messages in the sub data sets indeed have the desired properties. For the adaptive labels, all had the core term and 60% were not adaptive, fitting for meaning preserving. For refactoring, only 35% of the labels fit the meaning preserving data set, and in the corrective 75%. We use the same classifiers for the message and the summary, mitigating the accuracy importance. However, when comparing results between data sets, the classifier accuracy should be taken into account. Assuming independence, estimating mistakes with $P(Not\ Preserving) * (1 - Accuracy)$, which is much higher in corrective and adaptive compared to refactor.

The fact that meaning is often not preserved is likely to be general. We used off-the-shelf pre-trained models. Moreover, we did not train the model directly to preserve the meaning, rather we trained it to generate the summary token by token. Thus, the models are trained in a rather general way one that is not specific to summarizing commits or to preserving meaning. The only change is that the models were trained on data which meaning could be evaluated on. Thus, we rely on the distinctions known in software engineering research to evaluate the behavior of current models expecting it to be relevant to other summarization domains as well.

# 7 Ethical Considerations

The messages contained in the data set were written by 317,423 developers contributing to open source projects. We could not get their *direct* approval to use the messages in the data set. However, open source projects allow not only access to the commit messages but even to the source code. Developers are aware of that and agree to it as it is a part of the development project of all *public open source* projects. We validated in GitHub and all the projects included in ComSum have an OSI-approved open source license.

While we do not store developers' personal information, each commit is identified by a hash. Given the hash, a look up in the project metadata retrieves the developer's profile. Since it is required from the development process, the developers accept that and we do not ease look up or provide new information about the developer. In any case, the developer controls the data published on them and not us. Moreover, they can remove or alter it in any way that does not violate GitHub's terms. We consider this concern as addressed too.

Another concern is whether the data set is merely pointing out to commit messages as a possible source of text summarization. While this is one novelty of our work, it is not the case. Our work included filtering unsuitable projects, their vast majority. We also propose a dedicated evaluation procedure of meaning preservation. Hence, the value is larger than one would have gotten from just the idea of using commit messages.

Note that 7K commits were identified[7] to contain swearing and 325k commits were identified to contain negative sentiment. The true numbers might be higher due to the classifiers' false negatives. As this data is already open we did not filter those, but warn future users of the data to filter profanity if their needs so require.

# 8 Limitations and Threats to Validity

The data set is based on active open source projects. These projects will keep advancing and in the future will have more commits that will enable building larger data sets. We extract the current commits and freeze them to enable reproducibility. We also provide the extraction code and limit the commits to commits earlier than 2021. However, the current data set might not represent future development activity. A data set that will be generated in the future might not match the frozen data sets since projects that will be deleted will not be included.

For the meaning preserving analysis we use commit classification models. Each model has different biases and prediction performance. We believe that further improving current models and using more models will reduce the influence of current models weaknesses. In turn, this makes the model exact details part of the reporting and reproducibility.

As an external validity concern, it is not clear how much the projects in the data set represent open source development. While we control our project selection, we could not find documentation explaining how projects are selected into the BigQuery schema that we rely on. Some well known projects are not included (e.g. The Apache Software Foundation's Mahout and ActiveMQ). An absence which is even harder to explain is that of Microsoft's VSCode, an extremely popular editor with more than 100K stars. It existed and was later removed, though the project is still publicly developed. On the other hand, our data set contains 19,720 projects research more than is usual based on the GitHub schema: 7,557 Amit and Feitelson [2020], 1,531 Amit and Feitelson [2019], and 677 Amit et al. [2021].

Git enables developers to create commits in a dedicated 'branch' and then 'squash' them into a single commit. The default message of the squashed commit is the concatenated messages of all the branch commits. While all the commits in a branch are related, the cohesion is lower than in a regular commit and the messages are longer. These cases can be identified, either by filtering commits having more than one parent or simply long messages. We want to allow future researchers maximal flexibility and therefore we just alert on this issue instead of enforcing a specific filter.

Our data set is specific to software development. Hence, improvement on Comsum might not generalize to other domains. Clearly, bugs and refactoring will not appear in other domains. Other than

---

[7]Using classifiers from https://github.com/evidencebp/commit-classification

this obvious difference, high percent of software developers are males Terrell et al. [2017], raising another external validity threat.

Our choice to set a minimal difference of 100 characters between subject and message is not the only option. 48 million commits, 94% of all of the commits in our projects, have a message longer than their corresponding subject. Our choice led to an average ratio of $\frac{len(message)}{len(subject)} = 11.08$ , requiring significant compression. In this case we did not provide the messages with a smaller difference since that will require a much higher storage of less interesting or even misleading messages. The code that we provide enables others to generate similar data sets to their taste.

Training models is costly. Therefore we could not repeat the training on many samples in order to provide error bars for the benchmarks. However, we evaluate the performance on large test sets. Table 2 shows that both Bart, Zero-Shot Bart and 'Random Message Sentence' get very close results on the train and test which is another reason to believe results are robust.

# 9   Future Work

The commit data set has the important property of task type meaning-preserving. This property enables requiring and evaluating beyond lexical similarity. It will be interesting to identify such properties in general texts. For example, forbidding a change from a positive sentiment to a negative one (e.g., in dropping the 'not' in 'not bad') might be a general property. Negation, modals, and idioms seem to be a suitable area to find such properties. Text topics, like security or performance in commit messages, might be suitable for meaning preserving too.

Our data set is also useful as a test bed for active learning. In active learning, there is a large amount of unlabeled data and the goal is to find a small group of informative samples that can be labeled in a feasible cost Settles [2010]. One can use labeling functions [Ratner et al., 2016, Amit et al., 2017], computational functions that are weak learners [Schapire, 1990]. For example, commits not classified as neither corrective, perfective or adaptive, are assured to be a false negative of one of the classifiers. The method was used to boost the corrective commit classifier model [Amit and Feitelson, 2020].

One can use the 19,720 projects for topic detection, a data set that is expected to be challenging since all the projects deal with software and hence the topic difference is more delicate. Another possible use is to enhance the data set with author identifier, and use pairing [Amit et al., 2019] in order to learn author writing style.

# 10   Conclusions

We present a text summarization data set, ComSum, of significant size, and a methodology to extract larger such data sets in the future. ComSum is not only of a large size, it provides new challenges such as summarizing in a new domain, where a lot of terms appear and constantly change.

We present benchmarks based on related messages, allowing us to assign meaning to a model performance evaluation. We also identify meaning-preserving properties that enable training and evaluating models on goals beyond lexical similarity.

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

## A Supplementary Materials

Source code and documentation are available at Choshen and Amit [2021b] and `https://github.com/evidencebp/comsum`. Data is available at Choshen and Amit [2021a].

## B Labeling for the administrative heuristic

Commit message can be viewd as contanining content describing the code change (e.g., 'extracted method') and administrative content (e.g., 'Signed-off-by: Alan Turing').

The administrative content usually uses few specific terms that can be identified. Our heuristic looks for these terms in the message and classify it as administrative if an administrative term appears in the first 20 characters. The intuiting of the heuristic is that 20 characters do not leave space for code change description. We labeled all 265 hits in a 5,000 commits samples. 2 samples were summaries with a change/details relation. 1 sample was a merge. 98.9% of the matches needed removing. 46 samples, with distance closer to 20 were reference commits (e.g., fixed bug #123). These are also not suitable for summary and should be removed. The hit rate is 5.3% compared to 7% positive rate

in the random sample, indicating a recall of about 75%. All administrative commits in the random samples were identified.

## C  Author Statement of Responsibility

We, the authors of "ComSum: Commit Messages Summarization and Meaning Preservation" bear all responsibility in case of violation of rights, etc. due to the publication of the data set.

We publish the data set with the license Creative Commons version 4.0 (aka, CC-4) in order to enable researchers to use it.

## D  Hosting, Licensing, and Maintenance plan

We release the data with the license Creative Commons version 4.0 (aka, CC-4), allowing copy, redistribution and other common uses without the need of permission but with proper credit. We do not plan to change it.

For the purpose of reviewing we host the data set in figshare. After publication, we will host the data set at GitHub. Sharing in GitHub has a built-in modification and tracking mechanism. This way, it is easy to add clarification, utility code, etc. Other than that, GitHub is the ideal hosting service for a data set of GitHub commit messages.

As for the maintenance plan, we provide all the code used to generate the data set. The infrastructure code is already public and it is not linked currently to preserve anonymity. Of course, it will be linked after publication.

The code will enable any researcher to maintain the data set and keep extending it. This includes adding data from future work in the projects, using different selection conditions, enhancement with more features, etc.

