# OpenReview forum: "ComSum: Commit Messages Summarization and Meaning Preservation"
_NeurIPS.cc/2021/Track/Datasets_and_Benchmarks/Round1 — Submitted to NeurIPS 2021 Datasets and Benchmarks Track (Round 1)_

### Official Review · Reviewer_6E8i · 2021-07-01
**Why subjects should be summary of commits?**

**Rating:** 4
**Confidence:** 4

**Strengths:**

- The main contribution of the work is they crawled the data and do filtering/cleaning of such data.

**Weaknesses:**

- The hypothesis that "title/subject is the summary of the commits" is not very motivated and persuasive.
- Very weak data analysis and comparison to existing work, for examples, what is the ratio that title is included in the commits? what is the min, average, max number of words (not charactors) in titles and in commits? What is the overalapping and similarity among commits in the same post?
- No qualitative study. The data seems to be quite noisy in my opinion when I check a few of them manually.
- Experiment-wise, it is not quite clear that what is the relationship between commits. For example, do we necessarily need to use all the commits to generate the title, or maybe just 1-2 commits are already fine.


**Additional Feedback:**

N/A

**Clarity:**

- There are many aspects that the paper can be improved. For examples, many paragraphs in the paper are too short (e.g., line 19-20, line 23-25, line 108-110).
- Spending too much space talking about cleaning and filtering, which is less important than the data analysis itself.

**Correctness:**

- I have doubts about the quality of the dataset as a summarization task.


**Documentation:**

- I would say not very comprehensive, scoring 3 out of 10.


**Ethics:**

- N/A, the data crawled are the publicly-available repositories.

**Relation To Prior Work:**

- Weak comparison to previous summarization dataset work.

**Summary And Contributions:**

The authors crawled code commits from Github and reformed it as a text summarization task.

---

> ### Author Response · Authors · 2021-07-08
> **previous summarization dataset work**
>
> We would kindly ask for references for work you deem unreferenced. We note the places where such works were mentioned such as in the related work sections at the beginning of the paper (lines 29-65).

---

### Official Review · Reviewer_CqJD · 2021-07-04
**Paper about large scale summarization dataset requiring some polishing.**

**Rating:** 6
**Confidence:** 3

**Strengths:**

The dataset presented is novel and seems freshly collected -- the dataset makes a contribution to a space lacking a dataset like the one presented here. The idea of meaning preservation for the summaries is also interesting and novel.


**Weaknesses:**

The main limitation of the work to me is the dataset splits for train, development, and evaluation. The current dataset makes a random split based on the commit hash, i would recommend making time and project sensitive splits instead. Splits can be setup so that no commits from the same project overlap between the different splits, similarly commits from the past can be used as training data with the future serving as test.

Update post discussion period: The authors promise addition of a train/test split and experimental results with no leakage across splits.

**Additional Feedback:**

NA.

**Clarity:**

The paper is largely well written although some of the sections could benefit from some consolidation. For example:
Section 5, L162-186: Consider listing and describing the "baselines" in a latex description environment.
Section 6: It would be helpful if "concept", "term", "core term" were used more consistently. Potentially also reducing the number of different terms used to refer to the same items.

**Correctness:**

The constructed dataset does not seem to have undergone any human validation in its construction phase beyond a filtering of projects and commits. I would have liked to see a human evaluation of messages and summaries. If not the whole dataset then atleast of the test set alone.

Update post discussion period: The authors labelled ~100 message-subject pairs for whether the subjects were valid summaries and found ~80% of them to be summaries, they also state that filtering can allow up to 90% of pairs to be summaries. This is as a reasonable initial exploration of how clean the dataset is. These levels of noise though not a problem for training are likely a problem for reasonable evaluation. A manually filtered test/dev set to facilitate clean evaluation would be a useful contribution of this work.


**Documentation:**

The dataset should have included a datasheet (https://arxiv.org/abs/1803.09010) providing all documentation on the dataset.

Update post discussion period: The authors have added a datasheet.

**Ethics:**

No.

**Relation To Prior Work:**

Well placed with respect to broader literature.

**Summary And Contributions:**

The paper releases an automatically constructed dataset for commit message summarization. The dataset pairs a long form commit message on github with a summary of the same message with the goal to generate the summary given the message.

---

> ### Author Response · Authors · 2021-07-07
> **Splitting the data set by projects**
>
> Thank you for the idea. I liked it very much.
>
> I added a split of projects in a similar way to the commits.
> I applied a hash function on the project name so the split is reproducible.
> I will add the split to the supplementary materials.

---

> ### Author Response · Authors · 2021-07-07
> **Human evaluation**
>
> We manually labeled thousands of commits, which can be found here: https://github.com/evidencebp/commit-classification
>
> 95% of the commits have enough information that enables different annotators to reach agreement on their classification (See https://arxiv.org/abs/2007.10912 journal version accepted and to appear soon).

---

> ### Comment · Reviewer_CqJD · 2021-07-09
> **Human evaluation of summaries and datasheet.**
>
> Thank you for attending to the requests from the review. Im glad you added the datasheet and the splits based on the projects.
>
> Some comments:
> - I would also recommend recomputing experimental results with the updated splits. I think having project commits across train and dev will likely lead to an overestimate of performance. In future versions of the paper it would also be interesting to see error analysis based on specific commit examples or a whole projects commits (perhaps different orgs with the repos have different standards for their commit messages?) indicating the specific challenges posed by this dataset.
> - I am not sure how the pointer to the work on how annotators reach agreement on commit classification (https://arxiv.org/abs/2007.10912) helps this work - perhaps this is about the labels used for meaning preservation. The 100 examples you labelled are in line with my request. I would think that it is worrying that 20% of the cases are classified as not summaries, while examining 100 samples is reasonable as a pilot I would encourage a more thorough human evaluation where substantially more examples are manually labelled by multiple annotators and that agreement metrics on this be reported. It would be ideal if the human labelled set serves as a truly gold test set. This could be in addition to a automatically gathered test set and other challenge sets for evaluation.
> - I would consider having 10% of your data being is noisy after rule based filtering as un-ideal. If you decide to only heuristically filter examples I would recommend very explicitly documenting which specific kinds of filters were applied and supplying this alongside your dataset for every example in the dataset so future analysis can be informed by this.
> - I would think that human examination will also allow you to better filter commits themselves. Allowing you to filter out cases of the kind highlighted by Reviewer 3 (6E8i): "commits that share the same title.".
> - I appreciate you adding a labeling protocol. This is important and I would recommend adding examples to this document if you decide to go the route of labeling a much larger test set. As it is, to a non-annotator, it is hard to understand what each guideline means.
>
> In general I also agree with most of Reviewer 3s (6E8i) comments. Specifically: "why your annotations are informative, where can we apply your task to help in downstream tasks, how difficult and reasonable the data and task is if you ask human to do that, etc." -- Adding potential downstream applications or arguments of significance, providing estimates of data quality, and task difficulty will make this a substantially more valuable resource to the community.

---

### Official Review · Reviewer_o5PW · 2021-07-04
**Large-scale dataset for text summarisation derived from commit messages**

**Rating:** 6
**Confidence:** 3
**Clarity:** The paper is well written and has a c…

**Strengths:**

The main strengths of this paper are that it introduces a novel and very large dataset for the text summarisation task in the software engineering domain. The addition of ethical considerations and potential limitations of the study are a good addition to the paper.

**Weaknesses:**

The main limitation of this paper is that the dataset contains data form a niche domain and it is questionable what is the usefulness of the dataset for building text summarisation models outside of this domain.

**Additional Feedback:**

All in all I liked the paper very much and it introduces a nice new dataset for the text summarisation task. I worry for the usefulness of the dataset as it contains data from a very narrow domain.

**Correctness:**

For the best of my knowledge the dataset has been constructed in a correct and sound way.

**Documentation:**

The paper contains sufficient information about the data collection method and approach. The addition of the data and code in the supplementary material are a good addition.

**Ethics:**

The authors discuss the potential ethical concerns. The main concern is that it has not been possible to ask consent from the developers who have submitted the commits to the open source projects and although the developers usernames have been removed it is in principle possible to identify the users from the commit hashes. I am not an expert in ethical issues and I am not sure if this poses a problem for the current work, but in my view the discussion of these issues in the paper is sufficient.

**Relation To Prior Work:**

The related work section contains sufficient and relevant references.

**Summary And Contributions:**

The paper introduces a very large dataset for text summarisation purposes derived from GitHub. The dataset contains the commit title and the description text of commits and the idea is that the title represents a summary of the description text. The main contributions of this paper are that the dataset if very large compared to other text summarisation datasets and also that it is specific to software engineering domain. For these reason it is a novel dataset and a useful addition to the field. However, the fact that the dataset has been derived from commit messages limits its usefulness outside of this domain.

---

> ### Author Response · Authors · 2021-07-08
> **data form a niche domain**
>
> Thank you for the thoughtful review,
> We generally agree that this data is of a specific domain, but wish to emphasize that if summarization systems succeed in several different domains, it is a strong sign of their usefulness (as in real life one would probably care for another specific domain), hence datasets of diverse domains are an advantage.
>
>  Also, you noted the dataset size, having a larger dataset could help in finding whether and where data stops helping. If it keeps on helping we expect it to serve as a large boost when mixed into pretrained models labeled data aiding low resource summarization.
>
> All of this is of course only motivational, as the dataset itself is released in the hopes it will be found useful for future work. For an example where such data is already useful, note “the pile”(The Pile: An 800GB Dataset of Diverse Text for Language Modeling) GPT-J’s dataset, the opensource GPT3 recreation. This dataset mixes several domains: Wikipedia, arXiv, GitHub, StackExchange, PubMed and HackerNews.
>
> On a side note, while news datasets may deal with broader topics, their title generation is still a specific domain in terms of the requirements from a summarization system.

---

### Decision · Program_Chairs · 2021-07-26

**Decision:**

Reject

**Comment:**

Reviewers agree that this is an interesting topic but worry about the utility of the dataset is limited and also has some ethical issues. Also, it will require a series of additional work to make it ready for publication, for example:
(1) how to justify: "title/subject is the summary of the commits"
(2) more data analysis
(3) human evaluation of dataset's quality.
I’m inclined to reject the paper.